# A Qualitative Meta-Synthesis of Studies on Workplace Bullying among Nurses

**DOI:** 10.3390/ijerph192114120

**Published:** 2022-10-29

**Authors:** Haeyoung Lee, Young Mi Ryu, Mi Yu, Haejin Kim, Seieun Oh

**Affiliations:** 1College of Nursing, Chung-Ang University, Seoul 06974, Korea; 2Department of Nursing, Baekseok University, Cheonan 31065, Korea; 3College of Nursing, Institute of Health Sciences, Gyeongsang National Universtiy, Jinju 52727, Korea; 4Department of Nursing, Changwon National University, Changwon 51140, Korea; 5College of Nursing, Dankook University, Cheonan 31116, Korea

**Keywords:** workplace bullying, horizontal violence, lateral violence, clinical nurses, qualitative meta-synthesis

## Abstract

This study aimed to further understand and compare the phenomenon of workplace bullying (WPB) among clinical nurses in various sociocultural contexts. The study sought to determine appropriate interventions, examining how said interventions should be delivered at individual, work-unit, and institutional levels. Qualitative meta-synthesis was chosen to achieve the study aims. Individual qualitative research findings were gathered, compared, and summarized using the thematic analysis suggested by Braun and Clark. Based on the predefined analytic points, the findings included the following themes: horizontal yet vertical violence, direct and indirect violence on victims, nurses feed on their own, accepting and condoning WPB embedded in ineffective work systems, and rippling over the entire organization. The results showed that the phenomenon of workplace bullying shares quite a few attributes across cultures in terms of the characteristics, types, perpetrators, subjects, and consequences. The findings suggest that interventions to change and improve organizational work culture must be developed and implemented.

## 1. Introduction

Workplace bullying (WPB), or horizontal violence among healthcare workers, is a global and cross-cultural phenomenon. Victims of such violence suffer from long-term physical and psychological aftereffects, and they transfer from their departments, resign, or may even commit suicide in extreme cases. In addition, these long-term stressful situations reduce nursing competencies for patients, negatively affecting patient outcomes and significantly impacting hospital organizations’ productivity [1]. The seriousness of the consequences, including problems of individual employees being bullied, those of the organization as a whole, and harm to patients receiving care, mean that WPB must be resolved in healthcare organizations [2]. 

In Korea, harassment among nurses is a particularly well-known societal issue. The suicide incident of a nurse at a university hospital in 2018 [3], and a recent case of a nurse’s suicide when the workload of clinical nurses increased due to the COVID-19 pandemic [4], have highlighted the issue of harassment between nurses, which has again attracted social attention. Harassment among nurses in Korea is not new, and it has long been given special terms, such as “military culture” or “tae-um [Korean language].” In particular, “tae-um” is a representative term for harassment between nurses, which means “burning the soul until it turns to ash” and refers to the harassment that senior nurses inflict on new nurses in the name of education [5]. 

These incidents offer the following questions: Why does workplace harassment among nurses attract particular attention in Korean society? Is the degree of harassment in Korea particularly severe? What social and cultural factors in Korea increase WPB? What are the similarities and differences between WPB in Korea and WPB in other countries or cultures? To answer these questions, we analyzed and synthesized previous qualitative studies on the WPB phenomenon in Korea and other countries. Notably, this paper presents what we found regarding the last question of cross-national comparison. We synthesized qualitative studies so that we could grasp the contexts in which the research phenomena occurred within such studies and potentially understand their plausible explanations. The comparative analysis of qualitative studies on WPB occurring under various sociocultural backgrounds aimed to reveal the necessary interventions at the individual, unit, and institution levels required to provide and maintain safe environments for nurses and nursing recipients, and to obtain requisite information for the effective development and application of the interventions.

## 2. Methods

This is a qualitative meta-synthesis study, which uses the results of primary qualitative studies as raw data. Primarily, qualitative meta-synthesis (QMS) is based on literature review. However, QMS pursues new or greater understanding of a phenomenon of interest, than what can be found in individual qualitative research, by analyzing and synthesizing the raw data from individual qualitative research [6,7]. Since our study aimed to determine potential underlying mechanisms, or conditions, under which WPB may be expressed differently, and any commonalities or differences in WPB across countries, we chose to employ QMS to achieve our goals.

QMS methods undertake five common steps: formulating research questions, retrieving relevant qualitative research studies, appraising the quality of selected studies, analyzing and synthesizing the results, and securing the validity of the process and synthesized findings. The QMS approach requires analytic or synthetic methods to integrate the findings from primary qualitative studies. The terminology, QMS, itself does not refer to a specific method or technique [8] and the purpose of a study, the type of desired output, and the characteristics of individual research results to be included in the QMS should be the foundation for method selection [9] (pp. xv–xix, 1–10). 

This study used the analytic points pre-defined by the researchers to compare and contrast WPB among nurses, occurring under various social and cultural influences, to further understand the phenomenon. Therefore, Braun and Clarke’s thematic analysis [10], an analytic and synthetic method that allows this approach, was selected as the analysis and integration approach to obtain the results.

### 2.1. Search Methods

This study included qualitative studies on the WPB experiences of nurses for comparative analysis to deeply understand WPB among nurses working in medical institutions, and to compare the situation in Korea with those of other countries. Therefore, studies on conflicts or difficulties among mixed healthcare professionals, such as midwives, doctors, and nursing assistants were excluded as it was difficult to separate out nurses’ experiences. Non-research papers or review papers were excluded because they contained no raw data that could be used in this study. Studies not written in either Korean or English were also excluded as those are the only languages in which this study’s authors are fluent. In addition, to enable the collection of rich and comprehensive raw data on the research phenomena, we placed no restrictions on the period or research methods. For mixed-method studies, the main texts were reviewed to determine the applicability of the qualitative data, and inclusion was decided based on discussion within the research team. 

The process of literature retrieval was as follows. For literature in Korea, the electronic databases of the Research Information Sharing Service (RISS), the Korean studies Information Service System (KISS), the DataBase Periodical Information Academic (DBPia), the National Digital Science Library (NDSL), and the National Assembly Library were searched. The RISS contains academic references produced and owned by Korean universities, whereas the KISS offers bibliographic information published by academic societies and research institutes in Korea. The DBPia and NDSL are academic information portals that contain research studies published in Korea. All of the databases we used are the most searched databases and academic resource portals and have slightly different ranges of academic disciplines, types of resources, and systems of organizing resources. Literature published until June 2020, the final search period in Korea, was included. The search criteria phrase for the databases was “nurse” AND “qualitative” AND “(burning OR harassment OR violence OR hospital culture OR organizational culture OR burnout OR turnover OR resignation)”. 

For foreign literature, EBSCO, PubMed, CINAHL, Web of Science, EMBASE, and PsycINFO were searched. Foreign literature published until June 2020, the final search period of the literature search in Korea, was included and the search query for the databases was “nurs* AND (incivilit* OR bully* OR workplace violence OR uncivil OR aggression* OR harass*) AND (hospital* OR clinic* OR workplace*) AND (qualitative* OR phenomenological* OR grounded* OR grounded* OR (ethnographic*)”.

First, duplicate literature was removed, and then the title and abstract of each study was reviewed to confirm whether the paper met the inclusion criteria. If it was difficult to judge whether a study fit the criteria only by the title or abstract, the contents and results of the papers were read to determine whether the literature met the inclusion criteria. Then, the main texts of the selected studies were reviewed, and the literature to be included in this study was finally selected. The literature search and selection were performed independently among the researchers, but the selected literature was cross-checked in several research meetings. If there were disagreements during this process, the main texts were reviewed together to determine inclusion. After each of the authors reviewed the papers individually, we examined each paper together, based on our study aims, to decide whether the study under review contained “raw data” for our synthesis study. 

### 2.2. Search Outcomes

The literature search and selection were performed in four stages for Korea and other countries. In the Korean literature, 506 research papers were obtained from five electronic databases, of which 52 duplicate papers were excluded (Appendix A). Of the remaining 454 studies, 74 that were either not research papers or written in languages other than English or Korean were excluded. Of the 380 remaining papers, 314 studies that did not meet the inclusion criteria were excluded after reviews of their titles and abstracts. The excluded literature included studies unrelated to bullying among nurses, such as those about patient assault, those with subjects other than nurses, and surveys or intervention studies that did not use qualitative methods. Then, after reviews of the full texts of the remaining 66 papers, 31 studies were chosen for inclusion in the qualitative synthesis after 35 papers that did not meet the inclusion criteria, or for which the full text could not be retrieved, were excluded. After that, two related articles from the reference lists of other articles and four articles searched using the terms “rudeness” and “silence”, which were suggested as keywords from relevant literature, were added, resulting in 35 papers being included in the qualitative synthesis. 

For the literature from other countries, a total of 1995 research papers were obtained from six electronic databases, of which 1210 duplicate papers were excluded (Appendix A). Of the remaining 785 studies, 41 non-academic papers were excluded. After that, 664 of the 744 papers that did not meet the inclusion criteria were excluded, after reviews of their titles and abstracts. The excluded literature was made up of studies unrelated to bullying, those involving subjects other than nurses, surveys or intervention studies that did not use qualitative methods, and two studies that were not peer-reviewed. After the remaining 80 papers were reviewed, 52 studies that did not meet the research selection criteria, or whose original texts could not be found, were excluded from the analysis. Finally, 28 papers were included in the qualitative synthesis. Of these 28 included studies, the majority (13 studies) were conducted in the United States, followed by Australia (seven studies), and then one each for Iran, Chile, Turkey, New Zealand, Singapore, and South Africa.

### 2.3. Quality Appraisal

All 68 qualitative studies included in the synthesis were evaluated for their quality using the Critical Appraisal Skills Program (CASP) [11]. Among the 14 qualitative research quality evaluation tools used in the previous qualitative meta-synthesis studies, CASP was used here because it is the most commonly employed, it includes an evaluation category common to the 14 tools, it is relatively simple to apply, and enables the gathering of opinions. As the similarity of topics is more important than the quality of individual studies in qualitative synthesis, it was not appropriate to exclude studies due to quality issues [9]. Therefore, the quality evaluation results were used to improve the researchers’ understanding of individual studies and determine the reliability of the research results as raw data in this synthesis. No studies were excluded on account of their CASP score. Although there were slight differences in the quality of included studies, there was no study with such poor reliability that it could not be used for synthesis. 

### 2.4. Data Analysis and Synthesis 

The process of qualitative synthesis began with researchers familiarizing themselves with the contents of the included studies by reading them individually several times. Based on this, basic information about each study, including the purpose, characteristics of the research participants, data collection methods, and research methodology, were collected and summarized (Appendix A). Next, the results of the individual studies, comprising the raw data to be included in this study, were analyzed, while carefully reading and contrasting the topics or names of categories, the explanations or interpretations presented by the original authors and the research participants’ statements regarding each topic. After that, the results were recorded alongside the researchers’ interpretations. Based on these summarized results, we re-analyzed the studies by comparing and contrasting common findings from researchers, similarities and differences in results, and diversity among studies. Specifically, the comparison and analysis were repeated depending on the experiences of WPB revealed by individual studies, how the properties and forms of WPB appeared, the causes of WPB occurrence, factors that promoted or mitigated WPB, methods or situations that overcame WPB, and the results caused by WPB (Appendix A). After being repeatedly analyzed, data were classified, based on their similarity to other data, and the results with differences between studies were classified separately and re-reviewed to discern patterns or similarities. The content classified as similar was named using a concept that suitably expressed its nature. In addition, when the relationship between topics was analyzed, the findings were included in the synthesis results and described. As a final step, meta-synthesized topics, focusing on common results, were presented.

### 2.5. Optimization of the Study Validity 

The validity of the process and results of this qualitative synthesis study was ensured using the method proposed by Sandelowski and Barroso [9]. The first step was to record the activities conducted at each research stage, the decision-making process and the results as specifically as possible. While conducting this study, we collected and recorded various materials to ensure the reliability of the research process and results, such as synthesis results, the decision-making process and its results, reflective notes, tables, and figures derived through individual analysis and discussions between researchers. The second step was to secure negotiated consensual validity as a key procedure for securing validity in the qualitative synthesis. All decisions made throughout this study were considered for a sufficient amount of time, and the researchers tried to reach consensus through in-depth discussions. As several researchers participated, there were slightly varying opinions. By examining these differences through discussions, we could identify areas where the results were unclear or where there was a logical leap that warranted re-review. The differences of opinion were finally resolved by reviewing the original materials, including findings presented in individual papers and memos related to the findings.

## 3. Results

Thirty-five Korean and 28 international studies on nurses’ WPB experiences were analyzed and summarized by means of the following: The definition, attributes, causes, types, and perpetrators and victims of WPB; factors related to the occurrence of WPB; reactions to WPB; Consequences of WPB. In accordance with this study’s purpose, the comparison results were described in the corresponding areas if there was a difference between the Korean and non-Korean literature.

### 3.1. Horizontal But Vertical Violence: What Characterizes WPB?

WPB can be defined as verbal, physical, and emotional violence between nurses based on their work relationships within a nursing organization. Many studies have shown that this harassment has “deliberate”, “repetitive”, and “continuous” characteristics (K6, K8, K11, K31, K32, INT8, INT11, INT18, INT19, INT20, INT23, INT26), creates unfavorable or hostile working environments for victims of WPB, and is connived, passed down, and circulated within the organization (K28, K32, INT13, INT16, INT18, INT19, INT20). Based on these characteristics, WPB can be distinguished from simple one-off quarrels or interpersonal conflicts, and those involved in it easily expand to form groups (INT13).

Internationally, this violence has also been described as “lateral”, or “horizontal”, to characterize the context of violence committed between colleagues in the workplace. However, particularly in the Korean context, even if there is no difference in formal rank, it can be seen as vertical violence by hierarchy, rather than horizontal, because of the implicit, informal and authoritarian power hierarchy formed by differences in terms of years of experience in the ward. In a non-Korean context, being in the majority in terms of race, ethnicity, or age was a determining factor in the power hierarchy, and this characteristic sometimes overrode official rank. 


*A senior nurse made me take an exam in front of doctors and medical students because she thought I knew nothing. I was so embarrassed because she treated me ferociously with her eyes down and with scolding tone while I took the test in front of them. Once I made a mistake by misreading a pill label. I double-checked it with a nurse who has 5-year experiences and she told me I did great on finding out the pharmacy’s mistake. But after a while, she called me out loud and said “Can’t you read the label? This is exactly the right one!” I looked at the label closely and found that I was the one who made the mistake. I told her I was so sorry, but she grabbed my collar around my neck. She poured her anger out to me because she heard a blame from a pharmacist. I felt so bad that I had a resignation interview.*

*(K32)*



*I was fearful of them. Because I thought that I was going to be in their firing line. I think there was, I think in my mind it looked like it had been happening for a long time and that, you know, the talk around the traps was, ‘Yes, I tried to do something about this and that, and this and that, and nothing ever happened or came of it’…You know you are in the firing line at that stage.*

*(INT16)*



*This bully would always try to see if she could take me down to another level and embarrass me in front of the entire group. It was always me at every single meeting, putting me down; criticizing anything I came up with.*

*(INT23)*


### 3.2. Direct and Indirect Violence toward the Victim: Types of WPB

Our critical analysis and comparison revealed that there are two different types of WPB, based on how it is carried out: direct and indirect. Direct violence includes behaviors that stand out in the face-to-face interactions between perpetrators and victims. While some verbal and physical violence is intentionally carried out in front of others, violence carried out in private or closed spaces remains invisible to others. Specifically, it includes verbal violence (K8, K11, K13, K20, K28, K31, K32, K33, K34, K35, INT4, INT6, INT13, INT14, INT17, INT18, INT21, INT28), physical violence (K8, K11, K13, K20, K28, INT6, INT14, INT17, INT18,), expressions of doubt or devaluation of work ability, surveillance (K3, K6, K7, K8, K20, INT15, INT19, INT21), intentional ignorance, ostracism or isolation (K6, K8, K10, K13, K20, INT8, INT13, INT14, INT18, INT21, INT28), intentional insult, embarrassment, hurting one’s pride (K7, K8, K10, K11, K20, K28), invasions of privacy, spreading rumors or gossip (INT14, INT4, INT5, INT17), and inappropriate comments unrelated to work (K8, K10, K28). As these actions are performed to control the victim, such as by breaking their spirit through actions or taming (K6, K11, K20), they are unilateral, lack consideration or respect for the victim, and are repeated until the perpetrator is satisfied.


*… I was in public and then taken to a warehouse and scolded. Of course, I was dragged out because I could not be scolded in front of the patient, but it was really tough for me… I think I really wanted to quit when I heard if my mom and dad know about how you are working, and what did you learn at school…*

*(K8)*



*The staff was threatening me that day, and all her friends ganged up against me when I reported it. Even though the person was transferred to another shift, her friends continued to give me a silent treatment causing me to be unhappy to come to work.*

*(INT8)*


Indirect violence is not limited to the relationship between perpetrators and victims; it negatively impacts the victim’s relationships with other ward members. Victims become marginalized, and the perpetrators exert their influence in the ward atmosphere. Examples of this include manipulating the working atmosphere, interfering with work (providing information, work assistance, creating a hostile and difficult working environment without providing medical equipment, K6, K8, K10, INT8, INT14, INT17, INT28, INT4, INT10, INT12), disadvantages at work (unfair assignment of shifts, patients, duties, responsibilities, working authority, K8, K10, K11, K12, K20, K30), hindering professional career development (promotion, salary, unfair evaluation, INT4, INT14, INT15, INT22), and interfering with legal responses to bullying (INT13, INT14). These individual and group actions injure the identity and self-esteem of the victimized nurses. 


*And I do not know that it is necessarily [that nurses] ‘eat their young’. I think it is ‘eat somebody that’s not part of the group’.*

*(INT15)*



*I have a friend in another ward, and every time I ask them, they say they work in the evening. Quoting what my friend said, it was really difficult because the work wasn’t fair… It’s about giving favorable shifts and assigning easy patients to only some nurses… It was really hard when I was scolded for not being able quickly to do my job while I had a lot of work to do… They didn’t mention important things to learn, asked to do things like chores, and didn’t teach important things just because they did not like them.*

*(K8)*


In Korean studies, forms of WPB that criticize “work competency” and “interpersonal relationships at work” were prominent, whereas non-Korean studies relatively more frequently reported experiences of personal and emotional harassment and being alienated or being forced to be a loner, in addition to the violence of attacking victims’ work ability, as in Korea.


*It was group behavior, with a nurse provoking other nurses. For example, when I enter the room in the morning and say ‘Good morning’, all of them suddenly leave the room.*

*(INT26)*



*I was totally alone…one patient in what I thought was SVT, one pulling out all of his lines because he was disoriented, and one who really seemed to have a hard time breathing. The RNs in the break room said they would be there “in a minute.” I called the supervisor [for help], and she told me to find my mentor. I was…all alone, all the time. Yet I was responsible.*

*(INT18)*



*Sometimes I used to feel that I wasn’t present there, and I wasn’t acknowledged when I was there for handovers and for any opinions regarding my patients.*

*(INT19)*


### 3.3. Nurses Feed on Their Own: Perpetrators and Victims of WPB

The WPB reported in the qualitative studies was repeated occurrence of violence by one or more individuals or groups within an organization, and had various causes, from individual tendencies to organizational characteristics. Although personal tendencies may play a role in becoming a victim or perpetrator of WPB, it is characterized by the abuse of unilateral power between the weak and strong, which is formally, or implicitly, created within the organization, due to the hierarchical order or imbalance of power. 

#### 3.3.1. Victims of WPB 

In both domestic and international studies, most of the victims of WPB were new nurses (K8, K11, K12, K13, K20, K17, K21, K22, K23, K24, K25, K26, K28, K30, K32, K33, K35, INT12, INT16, INT17, INT18, INT19, INT20, INT21, INT22, INT25). However, nurses with little hospital experience (K19, K20, K25), those with intermediate careers, preceptors (K30), or nursing managers (K13, INT26) were also targets of harassment. In other cases, ethnic minorities (INT15), a nurse (INT3) who was part of a minority group in the organization, and a young nurse (INT3) were also the targets of harassment. 

Heterogeneity between groups was also a factor that led to WPB: age (INT3, K16), place of education (K1), and race (INT3) were the criteria that divided groups. As nurse groups in Korea are mostly homogeneous (i.e., native Korean), no case of becoming a victim of WPB due to race or culture was seen.


*Hey, what does this abbreviation mean? What school did you go to? Did your school teach you that way?*

*(K29)*



*And then race got into play. I am Hispanic, but many people on the floor think I look Muslim. And then …. a co-worker approached me and asked me if I can take less assignments because they thought I was Muslim and they didn’t feel comfortable with me working with them, with other—with their patients as well.*

*(INT15)*


#### 3.3.2. Perpetrators of WPB 

The most common perpetrators of WPB were fellow nurses who had longer careers (K11, K12, K13, K15, K20, K22, K23, K24, K26, K30, K31, K32, K33, K35. INT1, INT2, INT4, INT5, INT6, INT10, INT11, INT16, INT17, INT19, INT20, INT21, INT22, INT23, INT24, INT26, INT27, INT28). They included preceptors (K21, K25, K30, K33, INT25), charge nurses, nursing managers (K11, K12, K33, INT2, INT4, INT18, INT19, INT20, INT22, INT25, INT26, INT27, INT28), or nursing department heads (K11), and, in some cases, even White and non-immigrant nurses (INT3). In other words, various types of perpetrators can exist, depending on the relationship with the victim within the organization, but perpetrators were people who had an advantage within the hierarchy of the organization. Bullies need to dominate, create conflict to exert power over violence, and have no fear of repercussions for continuing their harassment of victims (INT1). In addition to these personal characteristics, they form a bullying hierarchy as an informal network promoting WPB (INT13). In addition to organizational culture and work characteristics, individual issues of the perpetrators were also causes of WPB. Examples include professional jealousy of others, fatigue, anger, insecurity, and being hateful (INT2), dissatisfaction, jealousy, and prejudice (INT26); and personal situations of nurses (lack of self-care, imbalance between home and work, etc.) (INT17).


*There seems to be an aspect of relieving one’s own personal stress and a backlash to the subordinates who did not respect properly…… It seems like ‘I retaliate because you ignored me’.*

*(K12)*



*I think we’ve got some very incestuous relationships here in senior executive. I think it’s very hard for people, particularly if they’re outsiders that haven’t grown up here, gone to school here, trained with everybody, worked with everybody for the last twenty, thirty years, all those people have moved up into higher positions. They’ve got a vested interest in keeping people where they are.*

*(INT13)*


### 3.4. Accepting and Condoning WPB Embedded in Ineffective Work Systems: What Makes WPB Long-Lasting?

Several studies in Korea and in other countries reported the power structures caused by inefficient organizational culture (K1, K2, K8, K11, K33, K34, K35, INT1, INT21, INT27) and power imbalance (K11, K31, K35, INT1, INT8, INT9) as being triggering factors for WPB. In an inefficient organizational culture, WPB occurs due to power imbalance. WPB was also explained as occurring in an “unfavorable environment” (INT18), which refers to systematic characteristics such as lack of various resources (lack of workforce, or lack of systematic education programs) (K13, INT26), lack of attention and competence (leadership) of a senior colleague (K30, INT1, INT22), and an improper organizational system (K33, K34, INT17)). These work environments amplify the situation and position of each class in the ward (head nurse, senior nurse, and new nurse) and cause conflict (K13). 

The major conditions that could sustain WPB within nursing organizations, in Korea and other countries, were the conformity of the victims and the connivance or apathy of colleagues and managers. The compliant attitude of victims of WPB (K6, K12, K13, K30) encouraged the perpetrators to continue their WPB. The perception of fellow nurses who viewed WPB as a part of adapting to work or the organization contributed to connivance with the perpetrators (K8, K28, INT8, INT18, INT20) and the organization members (manager or nursing department head) that knew about the victims or the ward’s atmosphere but just stood by and watched (K12, K13, K30, INT10, INT16, INT8, INT11, INT18, INT26), allowed the WPB to continue without obstacles.


*“Move on when it is about a person you know. Time will solve the problem. I have been through that much too. I have been through worse than you. Saying I cannot help, it is just okay. I was responding like this. They do not think seriously about this culture of bullying. No one thinks that bullying should be eradicated.”*

*(K13)*



*“When the director of the department didn’t respond, I went to senior leadership. However, they were friends with one another so no one would support me.”*

*(INT20)*


### 3.5. Personal Endeavor for Survival vs. Publicizing WPB: How to Deal with WPB

There were many similarities in how to respond to WPB in the studies analyzed. In the early stages, victims were embarrassed by the situation, greatly shocked emotionally, and intimidated, so they tended to withdraw into themselves, rather than confronting the perpetrators (K3, K6, K20, K17, K18, K29, K30, K31, K32, K33, INT4, INT11, INT12, INT15, INT16, INT26, INT27). Moreover, the tendency of WPB to be resolved through efforts to improve competencies to escape the ongoing WPB, or due to the improvement of practical skills as victims continue their careers, have been discussed in many studies worldwide (K13, K14, K27, K28, K29, K32, INT3, INT16, INT19). Cases of WPB reduction through the improvement of peer relations (K4, K5, K6, K13, K19, K27, K35, INT16, INT19) have also been commonly reported.

In studies of Korean nurses, victims had a clear tendency of considering themselves the cause of the problem because of nursing work requiring “continuity of work” through shifts and WPB frequently starting with accusations or conflicts related to “competency” (K6, K8, K12, K14, K27). Therefore, victims had a tendency to consider it as an unavoidable problem in patient care, or to recognize that individual survival efforts play a central role in changing the situation. 


*First of all, I think it is a process of making people suitable for the environment in order to adapt them to the organization. At that time, I was just annoyed and felt like ‘I can’t stand it anymore!’, but to a certain extent, I think that Tae-um [WPB] made me a good nurse.*

*(K12)*



*If we do not pay attention to our work, medical accidents happen and people die… I think this was the reason that it was connived even if it was a little excessive.*

*(K28)*


In contrast, confrontation with the perpetrator, or an attempt to formally resolve it, resulted in two different consequences in Korea and internationally. When the problem was reported to the manager or publicized, the problem was ignored (INT8), or victims were rather attacked, resulting in the situation getting worse (INT16, INT27); or they were further harmed by the organizational actions of bullies (INT12, INT15). Eventually, no further action could be taken, and they became silent (K6, K30, INT12, INT15, INT16, INT27). 


*“Why do you keep making excuses when it is unimportant who did it? I am just telling you not to do it. This was how I was scolded. So after that, I do not make excuses and say, ‘ah, I see, I am sorry’.”*

*(K6)*



*“I might be retaliated if I say this and that is a problem, and they say, “Look at her/him, she/he has such poor social skills”, and I could not say anything because I could get a worse disadvantage”.*

*(K30)*



*“I would say that I kept silent and tried to go through avoidance, that is, almost annulling myself and getting out of the way…”.*

*(INT4)*


Conversely, there were cases where WPB was mitigated or resolved when there was an active intervention at the administrator or organizational level (K13, K27, K32, INT2, INT3, INT4, INT8, INT12, INT15, INT17, INT20, INT24, INT25). In literature outside of Korea, solidarity among nurses, active response to WPB by managers or the entire nursing department, and active action or response by victims were recognized as important. 


*“We need to bring people together and find out what the root cause is and help one another to change”, “Maybe have early reporting of behaviors and commitment between nursing peers to address and stop this behavior”.*

*(INT20)*


It is essential to make it clear to their staff that nurse managers are ‘‘not going to tolerate this behavior.’’ Institution-wide ‘‘mandatory programs’’ were not deemed effective. Instead, nurse manager-initiated interventions on a unit level, in collaboration with institutional and administrative support, were perceived to be effective for addressing WPB among RNs (INT24). 

### 3.6. Rippling over the Entire Organization: WPB’s Consequences 

WPB negatively impacts nurses’ physical and mental health by causing depression, anxiety, posttraumatic stress disorder, insomnia, headaches, and indigestion (K3, K4, K6, K7, K8, K11, K20, K25, K27, K29, K30, K32, K35, INT1, INT4, INT6, INT9, INT11, INT19, INT26, INT27). If victims did not have a support system while experiencing WPB, they became more vulnerable to complete coercion and domination (INT10) and experienced self-contempt and frustration (INT11).


*“Whatever they tell me, I got dazed. I just kept losing motivation.”*

*(K4)*



*“I think it causes depression…And a lot of nurses are caregivers and they’re very emotional about their work and about taking care of people and they internalize that. And depression is anger turned inward. So you’re angry at the bully but instead of funneling back at them, you are turning it into yourself and that’s not healthy.*

*(INT1)*


The influence of WPB could extend beyond individual victims and spread to patients and organizations as a whole. In some cases, nurses in similar situations supported each other (K8) or tried to break the vicious cycle of the WPB situation (K6, K13, K19). However, WPB had negative effects on patient care, such as leading to burnout and eventually causing medical accidents such as medication errors or malpractice (K28, INT1, INT3, INT4, INT6, INT8, INT19, INT24). Eventually, the victims lost confidence in their nursing ability, lost their passion for their career, and burned out (K4, K31, K32, K33, K35, INT1, INT3, INT4, INT8, INT10, INT18, INT19, INT24, INT27). This negatively affected the quality of nursing care and the maintenance of nursing staff, resulting in long-term leave (INT10, INT13), absenteeism (INT3, INT20, INT21), resignation, or turnover (K5, K6, K14, K27, K32, INT1, INT3, INT7, INT9, INT13, INT15, INT16, INT19, INT20, INT21, A420, INT26). 


*“I pointed out the danger of assigning one nurse to monitor nine infants, two months and younger with RSV, without monitors and on tank oxygen. They expected the same nurse to cover additional patient orders. My nurse manager told me if I did not like it, I could leave.”*

*(INT8)*



*“… It is not just a matter of one department, but the whole hospital. As a result, experienced nurses also quit, and new nurses continue to come in… then there are more mistakes in practice… and the quality of nursing is getting worse and worse. In addition, when a new nurse quit, the reason for the resignation was driven by the problem of the remaining nurses while we were barely holding on to it. They tried to find out who the contributor was… and we became distrustful and blamed each other. It led to a complete loss of motivation.”*

*(K28)*


## 4. Discussion

In this study, we sought to determine the similarities and differences in WPB and the reasons behind them among nurses in Korea and those in other countries. Interestingly, we found more similarities than differences between Korea and other countries.

Characteristics and types of WPB were common across countries. Our findings showed that in both Korea and other countries, WPB has vertical as well as horizontal characteristics, due to unofficial power imbalance. Differences in these power imbalances seem to arise from social characteristics, (i.e., multicultural vs. monocultural society). Within multicultural societies, such as the U.S. or Australia, race, ethnicity, or characteristics of the majority in the nursing staff could provide covert power, whereas in Korea, the length of nursing career was the strongest factor for such invisible power.

In Korean studies, the length of one’s nursing career plays a significant role in the hierarchy. A plausible explanation for this can be found in the Korean national cultural character, which is highly hierarchical, resulting in the work environment in hospital settings also emphasizing hierarchy, which is based on the length of one’s nursing career. According to Hofstede’s model, Korea is highly hierarchical, so Korean people accept power imbalances by hierarchy with less resistance [12]. This tendency is also a possible explanation of Korean nurses’ acceptance of WPB as an unavoidable part of one’s adaptation to the job or job training. This underlying cultural influence seems to have allowed WPB in hospitals to be naturally contextualized as one of the ways to enhance work competency and reduce or prevent significant errors that might threaten the patients’ lives.

Considering the overt power hierarchy in the work unit or organization, the subjects of WPB were primarily the weakest in the hierarchy, young nurses. This was universal across all the countries where WPB studies were conducted. Additionally, a weaker position in the hierarchy could make anyone a victim of WPB, even experienced nurses or nurse managers, depending on the cultural circumstances. Perpetrators were those who had powers in any way and showed authoritative, pursuing, and wielding characteristics. Psychological characteristics of WPB victims and perpetrators were related to bullying behaviors. Homayuni et al. [13] reported that a high degree of core self-evaluations, such as self-esteem and self-efficacy, could have a preventive effect on WPB or be closely related to bullying behaviors.

The study results also illustrated close connections among inefficient organizational cultures, unfavorable work environments, and the amplification of work conflicts. Ineffective organizational culture creates and reinforces power imbalance, which, in turn, creates unfavorable work environments that are susceptible to WPB. Such environments encourage perpetrators’ continuation of WPB and make those who have less or no power more vulnerable to WPB. Under this context, responses to the WPB were also similar across various cultures. Psychological withdrawal and taking individual-level strategies to address WPB were common across countries. These tendencies are consistent with what Karatuna and colleagues [14] defined as “emotion-centered approaches” to resolve the issues, such as avoidance or seeking emotional support. Problem-centered approaches, such as speaking out and communicating directly with the bully or manager, were more prominent in studies conducted outside Korea. Although more attempts at confrontation were reported in international studies, even the pursuit of legal actions was often disrupted by WPB perpetrators.

Long-term consequences of WPB included negative impacts on individual victims’ physical and mental health, organizational productivity, and patient care outcomes. These results are consistent with those of many previous studies. For example, Shorey and Wong [15] reported that WPB negatively affects nurses’ physical and mental health; in addition, when a nurse becomes a target of WPB, their productivity and creativity decrease [15,16]. WPB’s adverse effect extended over patient safety [17] by decreasing the nurses’ competencies and quality of care, ultimately hindering the organization’s development [1,18,19,20]. Patients thus become the ultimate victims of WPB [18,21]. It has been reported that witnesses of harassment also experience stress, depression, and anxiety [15], which may cause them to experience similar consequences to direct victims [22]. 

Therefore, creating and maintaining safe, efficient work environments for nurses is one of the most urgent necessities for reducing, and ultimately eradicating, WPB in the field. Johnson [23], in his ecological model, emphasized the importance of a multi-layered and systematic approach to prevent bullying, claiming that orchestrated efforts should be implemented for successful prevention of WPB and the building of a collaborative, sound colleagueship and work atmosphere.

First, interventional approaches for individual nursing staff need to be considered. Institutional and unit-based supporting programs for new members’ adaptation and transitions are needed [24]. Such programs should contain training regarding work relationships and how to deal with conflicts with colleagues and WPB. In addition, it is also necessary to educate senior nurses, who can be WPB perpetrators, to distinguish between work rigor and harassment. Additionally, interventions to support senior nurses in managing work stress that they may experience, while educating or working with new nurses, need to be provided. As such, work environments that effectively reduce work conflicts will help prevent WPB as well [14]. 

As previous findings have shown, managerial or institutional intervention can be one of the most effective measures to deal with WPB [25]. Strategies that target nurse managers are needed to impart to them leadership skills that enable them to properly handle WPB. Improper leadership of nursing managers is highlighted as an aggravating factor causing WPB [14,16]. Therefore, there is a need for continuous education to ensure nursing managers recognize the negative effects of WPB on individuals and organizations and actively respond to it. Any measures that allow nurses to communicate with nurse managers or report bullying at any time, without feeling threatened or insecure, would enable immediate action in response to WPB [26]. 

At the institutional level, regular, systemic surveillance of WPB within organizations needs to be implemented. Not every situation under which WPB occurs and worsens can be handled by individual nurses or nurse managers. Accordingly, higher-level measures to monitor and effectively intervene WPB are necessary. Fortunately, due to the implementation of the Workplace Harassment Prohibition Act in 2019, all organizations in Korea are legally required to renew existing ethics committees, or launch such units, to formally report, manage, and prevent WPB. It was reported that organizations with a rigid, very vertical structure, job insecurity, and an adversarial and competitive work culture were more likely to report WPB [27]. On the contrary, organizations with guaranteed job security, clear expectations, and consistent rules had lower levels of WPB [16]. Political and relational conflicts amplify WPB in nursing organizations [28]. Active efforts should, thus, be made to secure an adequate nursing workforce, improve the work environment and solve the problems of nursing workforce allocation, supply and demand, and turnover, caused by the shortage of nurses and poor working conditions, which are important causes of WPB. 

On a social level, effective regulation systems for WPB need to be operated so that such systems can have actual impacts on medical institutions. In the revised Labor Standards Act from July 2019 in Korea, a clause on the prohibition of WPB that stipulates the concept of WPB, prohibits it, and punishes offenders has recently been added (Article 76 Paragraph 2). However, implementing legal actions cannot resolve the issue; continuous social awareness and spontaneous controls over WPB are critical to eradicate WPB.

This study has the following limitations. First, it included only studies written in Korean and English, so it was not possible to compare WPB studies in non-English speaking countries. Second, although we tried to understand and compare WPB in Korea and other countries, the number of studies included from each country was very small, except for the United States and Australia. Therefore, it was impossible to analyze the WPB phenomenon in detail by culture. Third, the results are raw data of the results of the original articles, interpreted from the perspective of nurses who were the targets of WPB. Therefore, it is necessary to understand the research phenomenon from the perspective of nurses or nurse managers who act as active agents of WPB.

## 5. Conclusions

This study was conducted to reveal what, how, and why different types of WPB occur in Korea and in other national or cultural contexts in clinical nursing organizations. The findings showed more universality of the phenomenon across countries, in terms of characteristics, forms of the violence, targets and agents, conditions that triggered and sustained WPB, and responses to WPB. However, a distinctive feature of WPB in Korea was that it was strongly inculcated in nurses as an unavoidable part of job training and adaptation to the job. Improper power imbalances, reinforced by ineffective organizational work environments and systems, were common conditions that triggered and sustained WPB. Characteristics held by the majority (e.g., White and non-immigrants) could be the measure of unofficial power in other countries, whereas the length of one’s nursing career was the prominent measure of unofficial power in Korea. Eradicating WPB in nursing fields is mandatory to secure high-quality nursing care and, thereby, improve patient outcomes and safety. To achieve this goal, multi-level efforts should be implemented, including individual training on how to deal with WPB, unit level and organizational regulations, and national board-level regulations. Building collaborative colleagueship needs to be taught and encouraged in college-level education as well. 

## Data Availability

Not applicable.

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
