# Peer review of "A Qualitative Meta-Synthesis of Studies on Workplace Bullying among Nurses"

_ijerph, 2022, doi:10.3390/ijerph192114120_

Round 1
Reviewer 1 Report
Several observations are shared:
1. Please revise the following sentences to improve comprehension (syntaxis, relevance, or grammatical concerns). Line 69-72.
2. Expand on the process implemented to fix disagreements, Line 98-99.
3. Please revise for minor grammatical errors in Table S1.
Author Response
Point 1: Please revise the following sentences to improve comprehension (syntaxis, relevance, or grammatical concerns). Line 69-72.
Response 1: Thank you for raising this issue. We re-examined those sentences and revised them to enhance readability and properly convey what we wanted to through them. We had our revised manuscript edited by a professional native English-speaking editor. We believe that our revisions and the professional editing have improved the clarity of those sentences.
Point 2: Expand on the process implemented to fix disagreements, Line 98-99
Response 2: Thank you for raising this issue. We have elaborated the process for handling disagreements between our research team members (pp.3, line#113-116). We believe that this revision will help the readers to understand how disagreements were resolved.
Point 3: Please revise for minor grammatical errors in Table S1.
Response 3: As mentioned in Response 1, we have had all documents, including Table S1, edited by a professional native English-speaking editor to improve clarity and readability of our manuscript. We believe that our revisions and the professional editing have improved the readability of our paper and supplemental documents
Reviewer 2 Report
This is very important work, that affects nurses all over the world, not just in Korea. This work is long overdue and authors have done excellent job. Providing quotes from nurses about specific situation, provides important information how dire the situation is.
Under recommendations, I wondered if authors have considered, one of the options to be also reporting nurses bullying behaviors to Board of Nursing/Licensing Board and the board having rules/regulations in place to handle these types of cases as deterrent to workplace bullying. Nursing/Licensing Boards have in place mechanisms addressing patient complaints, unethical actions and behaviors by nurses etc. Instances of workplace bullying could be handled the same way, by Nursing/Licensing boards if no mechanisms are in place.
Another recommendation would be for hospitals/medical facilities to have mindfullness room/center for nurses and designate mandatory paid time for nurses to take off at least 20 minutes out of their work-day to relax.
Author Response
Point 1: This is very important work, that affects nurses all over the world, not just in Korea. This work is long overdue and authors have done excellent job. Providing quotes from nurses about specific situation, provides important information how dire the situation is.
Response 1: We appreciate such encouraging comments. We hope this study can be of help for those who are interested in reducing WPB and building constructive environments for nursing profession.
Point 2: Under recommendations, I wondered if authors have considered, one of the options to be also reporting nurses bullying behaviors to Board of Nursing/Licensing Board and the board having rules/regulations in place to handle these types of cases as deterrent to workplace bullying. Nursing/Licensing Boards have in place mechanisms addressing patient complaints, unethical actions and behaviors by nurses etc. Instances of workplace bullying could be handled the same way, by Nursing/Licensing boards if no mechanisms are in place.
Response 2: We appreciate this comment that it is important to eradicate WPB and build collaborative colleagueship among clinical nurses. We included this suggestion in the Discussion (pp.12, line#589-593) and Conclusion sections (pp.13, line#622-626).
Point 3: Another recommendation would be for hospitals/medical facilities to have mindfullness room/center for nurses and designate mandatory paid time for nurses to take off at least 20 minutes out of their work-day to relax.
Response 3: Thank you for suggesting this valuable idea to support nurses experiencing WPB. We included similar ideas in the Discussion section among institution-level interventions to support nurses (pp.12, line#561-568).
Reviewer 3 Report
The topic of workplace bullying is important. Although the authors set out with quite an ambitious goal, the execution is quite poor, I am sorry to say. This is an overview of what I believe are the most important flaws:
1. The description of the study design is vague at best, but probably not described at all. The study design is supposedly a literature review, but this is not described as such in the Methods. This is also reflected by the title. 'A qualitative meta-synthesis' could be a way of approaching the analysis, but it is not a study design by itself. Indeed, in the Methods section, qualitative meta-synthesis is explained under the header of 'Data analysis and synthesis' and not under 'Study design'. To take this point further, reference #7 appears to provide the main approach for the entire (literature) study. However, this is problematic, as this is a book chapter on how to synthesize qualitative research, and not on how to perform a literature study. In addition, I am completely in the dark why it is then also needed to use reference #8, about using thematic research in psychology. Both sources appear relatively old and not appropriate for supporting the (non-descript) study design. Indeed, it is telling that the section 'Study design' does not contain any references at all.
2. If a systematic literature is indeed attempted, this should preferably be performed and reported according to the PRISMA guidelines. As it is, the quality of comprehensive and transparent reporting is relatively poor. This makes it hard to evaluate the quality of the review.
3. Poor reporting is aggravated by the moderate level of English writing.
4. There are many fundamental inconsistencies, to start with regarding the research questions (which were apparently 'thrown' at the authors): On page 2, lines 45-48, the research questions are: "Why does workplace 45 harassment among nurses attract particular attention in Korean society? Is the degree of harassment severe, particularly in Korea? What social and cultural factors in Korea deepen workplace bullying (hereinafter referred to as "WPB")? What are the similarities and differences between WPB in Korea and WPB in other countries or cultures?" However, in the Discussion, very different questions are presented: "Why does WPB occur?" (p. 10); "What is the coping response of WPB?" (p. 10; I am not sure how WPB can have a coping response at all...); "What are the consequences and effects of WPB?" (p. 10) and "How can WPB be solved?" (p. 11). In addition, in the Results, yet other questions are presented, so in total three different set of research questions are described!
5. The authors claim on p2, lines 51-54 that "The reason behind synthesizing qualitative studies was that we could grasp the context in which the research phenomena occurred in qualitative studies compared to quantitative studies; moreover, it is easier to under-stand details that are difficult to understand through quantitative measurements." This is only partly true, and the original research question formulated in the introduction can also be answered by (reviewing) quantitative studies.
6. A final point I want to address is that the authors state they want to examine "he similarities and differences between WPB in Korea and WPB in other countries or cultures" (p. 2, lines 47-48). However, in the actual synthesis as described in the Results, all the included studies from Korea and other nations are taken together. So there is no cross-national or -cultural comparison at all. Of course, just merging all these studies in the synthesis, without further explanation, is very problematic.
Author Response
Point 1: The description of the study design is vague at best, but probably not described at all. The study design is supposedly a literature review, but this is not described as such in the Methods. This is also reflected by the title. 'A qualitative meta-synthesis' could be a way of approaching the analysis, but it is not a study design by itself. Indeed, in the Methods section, qualitative meta-synthesis is explained under the header of 'Data analysis and synthesis' and not under 'Study design'. To take this point further, reference #7 appears to provide the main approach for the entire (literature) study. However, this is problematic, as this is a book chapter on how to synthesize qualitative research, and not on how to perform a literature study. In addition, I am completely in the dark why it is then also needed to use reference #8, about using thematic research in psychology. Both sources appear relatively old and not appropriate for supporting the (non-descript) study design. Indeed, it is telling that the section 'Study design' does not contain any references at all.
Response 1: Thank you for pointing out the insufficient description and justification of the methodology. As many researchers consider meta-analysis as one of the study designs, we think that qualitative meta-synthesis (QMS) can also be an effective study design. As explained in the previous manuscript, the terminology ‘QMS’ itself does not refer to a specific method, and researchers can employ one of the methods for integrating qualitative research findings suggested (reference #7 is one of them) or take conventional analytic approaches to analyze and synthesize the findings, because raw data for QMS are also qualitative data from primary qualitative studies. Since selection of the QMS’ analytic approach depends heavily on the study purpose and outcomes, we employed the analytic frameworks presented in reference#8, as they can most effectively serve the best our intention in conducting this study. References #7 and #8 are classic works that are still widely used in healthcare research.
Since the reviewer critiqued the methodological rigor, we re-examined the entire Method section and found that the description of our choice of QMS potentially insufficient for readers who are not familiar with it to understand what the QMS is and how to conduct it. We removed the subheading, which might be confusing or problematic to some readers, and added explanations and justification regarding what QMS is, what it seeks, and common procedural steps we followed to conduct the current study (pp.2, line# 60-74). Also, we re-arranged some of the descriptions in the Data Analysis and Synthesis section into the overall description as the reviewer suggested in the beginning (pp.2, line72-75) and added an ‘Optimizing Validity’ subsection at the end of the Method section so that the readers can follow the process described in the overall description (pp.4, line# 180-196).
Point 2: If a systematic literature is indeed attempted, this should preferably be performed and reported according to the PRISMA guidelines. As it is, the quality of comprehensive and transparent reporting is relatively poor. This makes it hard to evaluate the quality of the review.
Response 2: Thank you for pointing out this issue. Since QMS is not identical to systematic review or meta-analysis, we modified the PRISMA flowchart (2009) to illustrate the actual literature search process and outcomes. Thus, the only difference between the supplementary figures we submitted (Fig. S1, Fig. S2) and PRISMA flowchart is our figures do not have an ‘additional records identified through other sources’ box, which was not applicable to our literature retrieval. Please refer to the supplementary figures.
Point 3: Poor reporting is aggravated by the moderate level of English writing.
Response 3: We are sorry that the level of English of the manuscript was not satisfactory. We actually used a professional editing by a native English editor before submitting the original version. We had our revised manuscript proof-read and edited to enhance clarity and readability this time, as well as submitting the certificate. We believe that the revised version is far superior and will be clear in meaning.
Point 4: There are many fundamental inconsistencies, to start with regarding the research questions (which were apparently 'thrown' at the authors): On page 2, lines 45-48, the research questions are: "Why does workplace 45 harassment among nurses attract particular attention in Korean society? Is the degree of harassment severe, particularly in Korea? What social and cultural factors in Korea deepen workplace bullying (hereinafter referred to as "WPB")? What are the similarities and differences between WPB in Korea and WPB in other countries or cultures?" However, in the Discussion, very different questions are presented: "Why does WPB occur?" (p. 10); "What is the coping response of WPB?" (p. 10; I am not sure how WPB can have a coping response at all...); "What are the consequences and effects of WPB?" (p. 10) and "How can WPB be solved?" (p. 11). In addition, in the Results, yet other questions are presented, so in total three different set of research questions are described!
Response 4: We appreciate your critique regarding inconsistencies between the Introduction and the Discussion in terms of research questions. The questions in the Discussion sections were the analytic points that we pre-determined to find similarities and differences of WPB between Korea and other countries. Thus, they did not “come out of nowhere” as the reviewer read, but we agreed with the reviewer that subheadings in the Discussion section could have confused readers. To reduce confusion and enhance readers’ understanding of the findings, we deleted the subheadings, revised and re-organized the Discussion section according to the similarities and differences described in the Results section (pp.10-12, line# 496-610).
Point 5: The authors claim on p2, lines 51-54 that "The reason behind synthesizing qualitative studies was that we could grasp the context in which the research phenomena occurred in qualitative studies compared to quantitative studies; moreover, it is easier to under-stand details that are difficult to understand through quantitative measurements." This is only partly true, and the original research question formulated in the introduction can also be answered by (reviewing) quantitative studies.
Response 5: We acknowledge that quantitative studies can provide rich information on various aspects of WPB; nevertheless, we do not think such information would give us sufficient answers for the questions we had because variables and their measurement tools are not the same, which makes them difficult to compare, and different study designs will significantly affect interpretation of the statistical findings. However, we acknowledge that the sentences in the Introduction that the reviewer pointed out may exaggerate the advantage of utilizing qualitative studies. Thus, we revised them with focus of why findings of qualitative studies need to be analyzed (pp.2, line# 49-53).
Point 6: A final point I want to address is that the authors state they want to examine "he similarities and differences between WPB in Korea and WPB in other countries or cultures" (p. 2, lines 47-48). However, in the actual synthesis as described in the Results, all the included studies from Korea and other nations are taken together. So there is no cross-national or -cultural comparison at all. Of course, just merging all these studies in the synthesis, without further explanation, is very problematic.
Response 6: We appreciate your critique regarding such combined analysis. Our intention was to reveal such similarities and differences across national or cultural contexts as detailed as possible. However, due to our limitation of language ability (Korean and English), it was not possible to include literature published in other languages, and except for in the U.S. and Australia, and only one study from each other country (Iran, Chile, Turkey, New Zealand, Singapore, and South Africa) was included. Thus, it was beyond our ability to describe cultural differences in detail for each study conducted in other countries. Instead, we explained the limitation of diversity of included studies in the end of the Search Outcomes (pp.3, line# 141-144) and Discussion sections (pp.12, line# 601-606).
Reviewer 4 Report
I think it is a very appropriate work in view of the current situation of tension in society.
I would like to see in the conclusions a comparison between the situation in Korea and the rest of the world.
Author Response
Point 1: I think it is a very appropriate work in view of the current situation of tension in society.
Response 1: We appreciate such encouraging comments. We hope this study can be of help for readers of the journal to see the universality of workplace bullying, as well as the subtle differences in terms of how it is taken place.
Point 2: I would like to see in the conclusions a comparison between the situation in Korea and the rest of the world.
Response 2: Thank you for your comment. We revised the conclusion by inserting comparisons between the situations in Korea and the other countries (pp. 13, line#614-622). We hope to be able to conduct a larger study through collaborations with researchers from other countries in the future.